# Network Randomization:
# A Simple Technique for Generalization in Deep Reinforcement Learning

**Kimin Lee**[1][*][†]**, Kibok Lee**[2][*]**, Jinwoo Shin**[1]**, Honglak Lee**[32]
[1]KAIST, [2]University of Michigan, [3]Google Brain

## Abstract

Deep reinforcement learning (RL) agents often fail to generalize to unseen environments (yet semantically similar to trained agents), particularly when they are trained on high-dimensional state spaces, such as images. In this paper, we propose a simple technique to improve a generalization ability of deep RL agents by introducing a randomized (convolutional) neural network that randomly perturbs input observations. It enables trained agents to adapt to new domains by learning robust features invariant across varied and randomized environments. Furthermore, we consider an inference method based on the Monte Carlo approximation to reduce the variance induced by this randomization. We demonstrate the superiority of our method across 2D CoinRun, 3D DeepMind Lab exploration and 3D robotics control tasks: it significantly outperforms various regularization and data augmentation methods for the same purpose. Code is available at github.com/pokaxpoka/netrand.

## 1 Introduction

Deep reinforcement learning (RL) has been applied to various applications, including board games (e.g., Go (Silver et al., 2017) and Chess (Silver et al., 2018)), video games (e.g., Atari games (Mnih et al., 2015) and StarCraft (Vinyals et al., 2017)), and complex robotics control tasks (Tobin et al., 2017; Ren et al., 2019). However, it has been evidenced in recent years that deep RL agents often struggle to generalize to new environments, even when semantically similar to trained agents (Farebrother et al., 2018; Zhang et al., 2018b; Gamrian & Goldberg, 2019; Cobbe et al., 2019). For example, RL agents that learned a near-optimal policy for training levels in a video game fail to perform accurately in unseen levels (Cobbe et al., 2019), while a human can seamlessly generalize across similar tasks. Namely, RL agents often overfit to training environments, thus the lack of generalization ability makes them unreliable in several applications, such as health care (Chakraborty & Murphy, 2014) and finance (Deng et al., 2016).

The generalization of RL agents can be characterized by visual changes (Cobbe et al., 2019; Gamrian & Goldberg, 2019), different dynamics (Packer et al., 2018), and various structures (Beattie et al., 2016; Wang et al., 2016). In this paper, we focus on the generalization across tasks where the trained agents take various unseen visual patterns at the test time, e.g., different styles of backgrounds, floors, and other objects (see Figure 1). We also found that RL agents completely fail due to small visual changes[1] because it is challenging to learn generalizable representations from high-dimensional input observations, such as images.

To improve generalization, several strategies, such as regularization (Farebrother et al., 2018; Zhang et al., 2018b; Cobbe et al., 2019) and data augmentation (Tobin et al., 2017; Ren et al., 2019), have been proposed in the literature (see Section 2 for further details). In particular, Tobin et al. (2017) showed that training RL agents in various environments generated by randomizing rendering in a simulator improves the generalization performance, leading to a better performance in real environments. This implies that RL agents can learn invariant and robust representations if diverse

---

[*]Equal contribution.

[†]Work done while at University of Michigan.

[1]Demonstrations on the CoinRun and DeepMind Lab exploration tasks are available at [link-video].

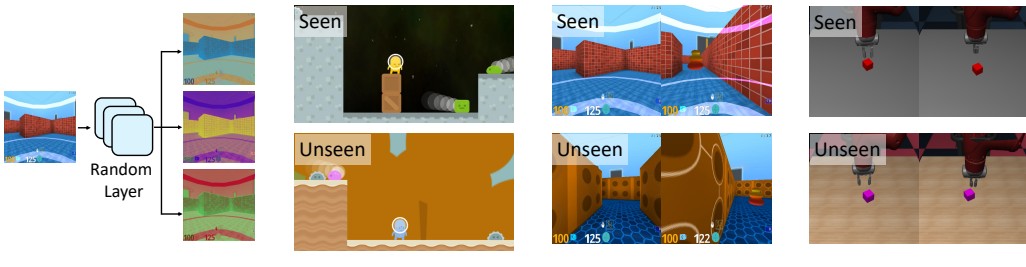

(a) Outputs of random layer     (b) 2D CoinRun     (c) 3D DeepMind Lab     (d) 3D Surreal robotics

Figure 1: (a) Examples of randomized inputs (color values in each channel are normalized for visualization) generated by re-initializing the parameters of a random layer. Examples of seen and unseen environments on (b) CoinRun, (c) DeepMind Lab, and (d) Surreal robotics control.

input observations are provided during training. However, their method is limited by requiring a physics simulator, which may not always be available. This motivates our approach of developing a simple and plausible method applicable to training deep RL agents.

The main contribution of this paper is to develop a simple randomization technique for improving the generalization ability across tasks with various unseen visual patterns. Our main idea is to utilize random (convolutional) networks to generate randomized inputs (see Figure 1(a)), and train RL agents (or their policy) by feeding them into the networks. Specifically, by re-initializing the parameters of random networks at every iteration, the agents are encouraged to be trained under a broad range of perturbed low-level features, e.g., various textures, colors, or shapes. We discover that the proposed idea guides RL agents to learn generalizable features that are more invariant in unseen environments (see Figure 3) than conventional regularization (Srivastava et al., 2014; Ioffe & Szegedy, 2015) and data augmentation (Cobbe et al., 2019; Cubuk et al., 2019) techniques. Here, we also provide an inference technique based on the Monte Carlo approximation, which stabilizes the performance by reducing the variance incurred from our randomization method at test time.

We demonstrate the effectiveness of the proposed method on the 2D CoinRun (Cobbe et al., 2019) game, the 3D DeepMind Lab exploration task (Beattie et al., 2016), and the 3D robotics control task (Fan et al., 2018). For evaluation, the performance of the trained agents is measured in unseen environments with various visual and geometrical patterns (e.g., different styles of backgrounds, objects, and floors), guaranteeing that the trained agents encounter unseen inputs at test time. Note that learning invariant and robust representations against such changes is essential to generalize to unseen environments. In our experiments, the proposed method significantly reduces the generalization gap in unseen environments unlike conventional regularization and data augmentation techniques. For example, compared to the agents learned with the cutout (DeVries & Taylor, 2017) data augmentation methods proposed by Cobbe et al. (2019), our method improves the success rates from 39.8% to 58.7% under 2D CoinRun, the total score from 55.4 to 358.2 for 3D DeepMind Lab, and the total score from 31.3 to 356.8 for the Surreal robotics control task. Our results can be influential to study other generalization domains, such as tasks with different dynamics (Packer et al., 2018), as well as solving real-world problems, such as sim-to-real transfer (Tobin et al., 2017).

## 2 RELATED WORK

**Generalization in deep RL**. Recently, the generalization performance of RL agents has been investigated by splitting training and test environments using random seeds (Zhang et al., 2018a) and distinct sets of levels in video games (Machado et al., 2018; Cobbe et al., 2019). Regularization is one of the major directions to improve the generalization ability of deep RL algorithms. Farebrother et al. (2018) and Cobbe et al. (2019) showed that regularization methods can improve the generalization performance of RL agents using various game modes of Atari (Machado et al., 2018) and procedurally generated arcade environments called CoinRun, respectively. On the other hand, data augmentation techniques have also been shown to improve generalization. Tobin et al. (2017) proposed a domain randomization method to generate simulated inputs by randomizing rendering in the simulator. Motivated by this, Cobbe et al. (2019) proposed a data augmentation method by modifying the cutout method (DeVries & Taylor, 2017). Our method can be combined with the prior methods to further improve the generalization performance.

**Random networks for deep RL**. Random networks have been utilized in several approaches for different purposes in deep RL. Burda et al. (2019) utilized a randomly initialized neural network to define an intrinsic reward for visiting unexplored states in challenging exploration problems. By learning to predict the reward from the random network, the agent can recognize unexplored states. Osband et al. (2018) studied a method to improve ensemble-based approaches by adding a randomized network to each ensemble member to improve the uncertainty estimation and efficient exploration in deep RL. Our method is different because we introduce a random network to improve the generalization ability of RL agents.

**Transfer learning**. Generalization is also closely related to transfer learning (Parisotto et al., 2016; Rusu et al., 2016a;b), which is used to improve the performance on a target task by transferring the knowledge from a source task. However, unlike supervised learning, it has been observed that fine-tuning a model pre-trained on the source task for adapting to the target task is not beneficial in deep RL. Therefore, Gamrian & Goldberg (2019) proposed a domain transfer method using generative adversarial networks (Goodfellow et al., 2014) and Farebrother et al. (2018) utilized regularization techniques to improve the performance of fine-tuning methods. Higgins et al. (2017) proposed a multi-stage RL, which learns to extract disentangled representations from the input observation and then trains the agents on the representations. Alternatively, we focus on the zero-shot performance of each agent at test time without further fine-tuning of the agent's parameters.

## 3 NETWORK RANDOMIZATION TECHNIQUE FOR GENERALIZATION

We consider a standard reinforcement learning (RL) framework where an agent interacts with an environment in discrete time. Formally, at each timestep $t$, the agent receives a state $s_t$ from the environment[2] and chooses an action $a_t$ based on its policy $\pi$. The environment returns a reward $r_t$ and the agent transitions to the next state $s_{t+1}$. The return $R_t = \sum_{k=0}^{\infty} \gamma^k r_{t+k}$ is the total accumulated rewards from timestep $t$ with a discount factor $\gamma \in [0, 1)$. RL then maximizes the expected return from each state $s_t$.

### 3.1 TRAINING AGENTS USING RANDOMIZED INPUT OBSERVATIONS

We introduce a random network $f$ with its parameters $\phi$ initialized with a prior distribution, e.g., Xavier normal distribution (Glorot & Bengio, 2010). Instead of the original input $s$, we train an agent using a randomized input $\widehat{s} = f(s; \phi)$.[3] For example, in the case of policy-based methods, the parameters $\theta$ of the policy network $\pi$ are optimized by minimizing the following policy gradient objective function:

$$\mathcal{L}_{\texttt{policy}}^{\texttt{random}} = \mathbb{E}_{(s_t, a_t, R_t) \in \mathcal{D}} \big[ -\log \pi \left( a_t | f\left( s_t; \phi \right); \theta \right) R_t \big], \tag{1}$$

where $\mathcal{D} = \{(s_t, a_t, R_t)\}$ is a set of past transitions with cumulative rewards. By re-initializing the parameters $\phi$ of the random network per iteration, the agents are trained using varied and randomized input observations (see Figure 1(a)). Namely, environments are generated with various visual patterns, but with the same semantics by randomizing the networks. Our agents are expected to adapt to new environments by learning invariant representation (see Figure 3 for supporting experiments).

To learn more invariant features, the following feature matching (FM) loss between hidden features from clean and randomized observations is also considered:

$$\mathcal{L}_{\texttt{FM}}^{\texttt{random}} = \mathbb{E}_{s_t \in \mathcal{D}} \big[ || h\left( f(s_t; \phi); \theta \right) - h\left( s_t; \theta \right) ||^2 \big], \tag{2}$$

where $h(\cdot)$ denotes the output of the penultimate layer of policy $\pi$. The hidden features from clean and randomized inputs are combined to learn more invariant features against the changes in the input observations.[4] Namely, the total loss is:

$$\mathcal{L}^{\texttt{random}} = \mathcal{L}_{\texttt{policy}}^{\texttt{random}} + \beta \mathcal{L}_{\texttt{FM}}^{\texttt{random}}, \tag{3}$$

where $\beta > 0$ is a hyper-parameter. The full procedure is summarized in Algorithm 1 in Appendix M.

---

[2]Throughout this paper, we focus on high-dimensional state spaces, e.g., images.

[3]Our method is applicable to the value-based methods as well.

[4]FM loss has also been investigated for various purposes: semi-supervised learning (Miyato et al., 2018) and unsupervised learning (Salimans et al., 2016; Xie et al., 2019).

| Method | Classification Accuracy (%) | |
|---|---|---|
| | Train (seen) | Test (unseen) |
| ResNet-18 | $95.0 \pm 2.4$ | $40.3 \pm 1.2$ |
| ResNet-18 + GR | $96.4 \pm 1.8$ | $70.9 \pm 1.7$ |
| ResNet-18 + CO | $95.9 \pm 2.3$ | $41.2 \pm 1.7$ |
| ResNet-18 + IV | $91.0 \pm 2.0$ | $47.1 \pm 15.1$ |
| ResNet-18 + CJ | $95.2 \pm 0.6$ | $43.5 \pm 0.3$ |
| ResNet-18 + ours | $95.9 \pm 1.6$ | $\mathbf{84.4} \pm 4.5$ |

Table 1: The classification accuracy (%) on dogs vs. cats dataset. The results show the mean and standard deviation averaged over three runs and the best result is indicated in bold.

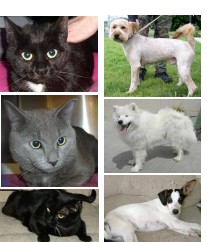 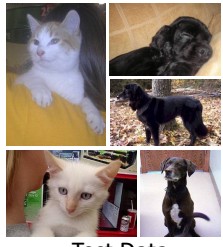

Training Data      Test Data

Figure 2: Samples of dogs vs. cats dataset. The training set consists of bright dogs and dark cats, whereas the test set consists of dark dogs and bright cats.

**Details of the random networks**. We propose to utilize a single-layer convolutional neural network (CNN) as a random network, where its output has the same dimension with the input (see Appendix D for additional experimental results on the various types of random networks). To re-initialize the parameters of the random network, we utilize the following mixture of distributions: $P(\phi) = \alpha \mathbb{I}(\phi = \mathbf{I}) + (1-\alpha)\mathcal{N}\left(\mathbf{0}; \sqrt{\frac{2}{n_{\text{in}}+n_{\text{out}}}}\right)$, where $\mathbf{I}$ is an identity kernel, $\alpha \in [0,1]$ is a positive constant, $\mathcal{N}$ denotes the normal distribution, and $n_{\text{in}}, n_{\text{out}}$ are the number of input and output channels, respectively. Here, clean inputs are used with the probability $\alpha$ because training only randomized inputs can complicate training. The Xavier normal distribution (Glorot & Bengio, 2010) is used for randomization because it maintains the variance of the input $s$ and the randomized input $\widehat{s}$. We empirically observe that this distribution stabilizes training.

**Removing visual bias**. To confirm the desired effects of our method, we conduct an image classification experiment on the dogs and cats database from *Kaggle*.[5] Following the same setup as Kim et al. (2019), we construct datasets with an undesirable bias as follows: the training set consists of bright dogs and dark cats while the test set consists of dark dogs and bright cats (see Appendix H for further details). A classifier is expected to make a decision based on the undesirable bias, (e.g., brightness and color) since CNNs are biased towards texture or color, rather than shape (Geirhos et al., 2019). Table 1 shows that ResNet-18 (He et al., 2016) does not generalize effectively due to overfitting to an undesirable bias in the training data. To address this issue, several image processing methods (Cubuk et al., 2019), such as grayout (GR), cutout (CO; DeVries & Taylor 2017), inversion (IV), and color jitter (CJ), can be applied (see Appendix C for further details). However, they are not effective in improving the generalization ability, compared to our method. This confirms that our approach makes DNNs capture more desired and meaningful information such as the shape by changing the visual appearance of attributes and entities in images while effectively keeping the semantic information. Prior sophisticated methods (Ganin et al., 2016; Kim et al., 2019) require additional information to eliminate such an undesired bias, while our method does not.[6] Although we mainly focus on RL applications, our idea can also be explorable in this direction.

### 3.2 INFERENCE METHODS FOR SMALL VARIANCE

Since the parameter of random networks is drawn from a prior distribution $P(\phi)$, our policy is modeled by a stochastic neural network: $\pi(a|s;\theta) = \mathbb{E}_\phi\big[\pi\left(a|f\left(s;\phi\right);\theta\right)\big]$. Based on this interpretation, our training procedure (i.e., randomizing the parameters) consists of training stochastic models using the Monte Carlo (MC) approximation (with one sample per iteration). Therefore, at the inference or test time, an action $a$ is taken by approximating the expectations as follows: $\pi(a|s;\theta) \simeq \frac{1}{M}\sum_{m=1}^{M}\pi\left(a|f\left(s;\phi^{(m)}\right);\theta\right)$, where $\phi^{(m)} \sim P(\phi)$ and $M$ is the number of MC samples. In other words, we generate $M$ random inputs for each observation and then aggregate their decisions. The results show that this estimator improves the performance of the trained agents by approximating the posterior distribution more accurately (see Figure 3(d)).

---

[5] https://www.kaggle.com/c/dogs-vs-cats
[6] Using the known bias information (i.e., {dark, bright}) and ImageNet pre-trained model, Kim et al. (2019) achieve 90.3%, while our method achieves 84.4% without using both inputs.

## 4 EXPERIMENTS

In this section, we demonstrate the effectiveness of the proposed method on 2D CoinRun (Cobbe et al., 2019), 3D DeepMind Lab exploration (Beattie et al., 2016), and 3D robotics control task (Fan et al., 2018). To evaluate the generalization ability, we measure the performance of trained agents in unseen environments which consist of different styles of backgrounds, objects, and floors. Due to the space limitation, we provide more detailed experimental setups and results in the Appendix.

### 4.1 BASELINES AND IMPLEMENTATION DETAILS

For CoinRun and DeepMind Lab experiments, similar to Cobbe et al. (2019), we take the CNN architecture used in IMPALA (Espeholt et al., 2018) as the policy network, and the Proximal Policy Optimization (PPO) (Schulman et al., 2017) method to train the agents.[7] At each timestep, agents are given an observation frame of size $64 \times 64$ as input (resized from the raw observation of size $320 \times 240$ as in the DeepMind Lab), and the trajectories are collected with the 256-step rollout for training. For Surreal robotics experiments, similar to Fan et al. (2018), the hybrid of CNN and long short-term memory (LSTM) architecture is taken as the policy network, and a distributed version of PPO (i.e., actors collect a massive amount of trajectories, and the centralized learner updates the model parameters using PPO) is used to train the agents.[8] We measure the performance in the unseen environment for every 10M timesteps and report the mean and standard deviation across three runs.

Our proposed method, which augments PPO with random networks and feature matching (FM) loss (denoted PPO + ours), is compared with several regularization and data augmentation methods. As regularization methods, we compare dropout (DO; Srivastava et al. 2014), L2 regularization (L2), and batch normalization (BN; Ioffe & Szegedy 2015). For those methods, we use the hyperparameters suggested in Cobbe et al. (2019), which are empirically shown to be effective: a dropout probability of 0.1 and a coefficient of $10^{-4}$ for L2 regularization. We also consider various data augmentation methods: a variant of cutout (CO; DeVries & Taylor 2017) proposed in Cobbe et al. (2019), grayout (GR), inversion (IV), and color jitter (CJ) by adjusting brightness, contrast, and saturation (see Appendix C for more details). As an upper bound, we report the performance of agents trained directly on unseen environments, dented PPO (oracle). For our method, we use $\beta = 0.002$ for the weight of the FM loss, $\alpha = 0.1$ for the probability of skipping the random network, $M = 10$ for MC approximation, and a single-layer CNN with the kernel size of 3 as a random network.

### 4.2 EXPERIMENTS ON COINRUN

**Task description**. In this task, an agent is located at the leftmost side of the map and the goal is to collect the coin located at the rightmost side of the map within 1,000 timesteps. The agent observes its surrounding environment in the third-person point of view, where the agent is always located at the center of the observation. CoinRun contains an arbitrarily large number of levels which are generated deterministically from a given seed. In each level, the style of background, floor, and obstacles is randomly selected from the available themes (34 backgrounds, 6 grounds, 5 agents, and 9 moving obstacles). Some obstacles and pitfalls are distributed between the agent and the coin, where a collision with them results in the agent's immediate death. We measure the success rates, which correspond to the number of collected coins divided by the number of played levels.

**Ablation study on small-scale environments**. First, we train agents on one level for 100M timesteps and measure the performance in unseen environments by only changing the style of the background, as shown in Figure 3(a). Note that these visual changes are not significant to the game's dynamics, but the agent should achieve a high success rate if it can generalize accurately. However, Table 2 shows that all baseline agents fail to generalize to unseen environments, while they achieve a near-optimal performance in the seen environment. This shows that regularization techniques have no significant impact on improving the generalization ability. Even though data augmentation techniques, such as cutout (CO) and color jitter (CJ), slightly improve the performance, our proposed method is most effective because it can produce a diverse novelty in attributes and entities. Training with randomized inputs can degrade the training performance, but the high expressive power

---

[7]We used a reference implementation in https://github.com/openai/coinrun.
[8]We used a reference implementation with two actors in https://github.com/SurrealAI/surreal.

| | | PPO | PPO + DO | PPO + L2 | PPO + BN | PPO + CO | PPO + IV | PPO + GR | PPO + CJ | PPO + ours Rand | Rand + FM |
|---|---|---|---|---|---|---|---|---|---|---|---|
| Success rate | Seen | 100 ± 0.0 | 100 ± 0.0 | 98.3 ± 2.9 | 93.3 ± 11.5 | 100 ± 0.0 | 95.0 ± 8.6 | 100 ± 0.0 | 100 ± 0.0 | 95.0 ± 7.1 | **100** ± 0.0 |
| | Unseen | 34.6 ± 4.5 | 25.3 ± 12.0 | 34.1 ± 5.4 | 31.5 ± 13.1 | 41.9 ± 5.5 | 37.5 ± 0.8 | 26.9 ± 13.1 | 43.1 ± 1.4 | 76.7 ± 1.3 | **78.1** ± 3.5 |
| Cycle-consistency | 2-way | 18.9 ± 10.9 | 13.3 ± 2.2 | 24.4 ± 1.1 | 25.5 ± 6.6 | 27.8 ± 10.6 | 17.8 ± 15.6 | 17.7 ± 1.1 | 32.2 ± 3.1 | 64.7 ± 4.4 | **67.8** ± 6.2 |
| | 3-way | 4.4 ± 2.2 | 4.4 ± 2.2 | 8.9 ± 3.8 | 7.4 ± 1.2 | 9.6 ± 5.6 | 5.6 ± 4.7 | 2.2 ± 3.8 | 15.6 ± 3.1 | 39.3 ± 8.5 | **43.3** ± 4.7 |

Table 2: Success rate (%) and cycle-consistency (%) after 100M timesteps in small-scale CoinRun. The results show the mean and standard deviation averaged over three runs and the best results are indicated in bold.

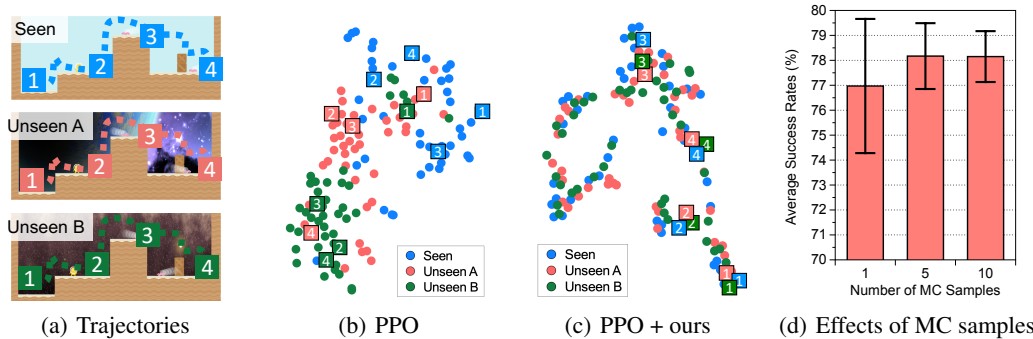

(a) Trajectories     (b) PPO     (c) PPO + ours     (d) Effects of MC samples

Figure 3: (a) We collect multiple episodes from various environments by human demonstrators and visualize the hidden representation of trained agents optimized by (b) PPO and (c) PPO + ours constructed by t-SNE, where the colors of points indicate the environments of the corresponding observations. (d) Average success rates for varying number of MC samples.

of DNNs prevents from it. The performance in unseen environments can be further improved by optimizing the FM loss. To verify the effectiveness of MC approximation at test time, we measure the performance in unseen environments by varying the number of MC samples. Figure 3(d) shows the mean and standard deviation across 50 evaluations. The performance and its variance can be improved by increasing the number of MC samples, but the improvement is saturated around ten samples. Thus, we use ten samples for the following experiments.

**Embedding analysis**. We analyze whether the hidden representation of trained RL agents exhibits meaningful abstraction in the unseen environments. The features on the penultimate layer of trained agents are visualized and reduced to two dimensions using t-SNE (Maaten & Hinton, 2008). Figure 3 shows the projection of trajectories taken by human demonstrators in seen and unseen environments (see Figure 17 in Appendix N for further results). Here, trajectories from both seen and unseen environments are aligned on the hidden space of our agents, while the baselines yield scattered and disjointed trajectories. This implies that our method makes RL agents capable of learning the invariant and robust representation.

To evaluate the quality of hidden representation quantitatively, the cycle-consistency proposed in Aytar et al. (2018) is also measured. Given two trajectories $V$ and $U$, $v_i \in V$ first locates its nearest neighbor in the other trajectory $u_j = \arg\min_{u \in U} \|h(v_i) - h(u)\|^2$, where $h(\cdot)$ denotes the output of the penultimate layer of trained agents. Then, the nearest neighbor of $u_j$ in $V$ is located, i.e., $v_k = \arg\min_{v \in V} \|h(v) - h(u_j)\|^2$, and $v_i$ is defined as cycle-consistent if $|i - k| \leq 1$, i.e., it can return to the original point. Note that this cycle-consistency implies that two trajectories are accurately aligned in the hidden space. Similar to Aytar et al. (2018), we also evaluate the three-way cycle-consistency by measuring whether $v_i$ remains cycle-consistent along both paths, $V \to U \to J \to V$ and $V \to J \to U \to V$, where $J$ is the third trajectory. Using the trajectories shown in Figure 3(a), Table 2 reports the percentage of input observations in the seen environment (blue curve) that are cycle-consistent with unseen trajectories (red and green curves). Similar to the results shown in Figure 3(c), our method significantly improves the cycle-consistency compared to the vanilla PPO agent.

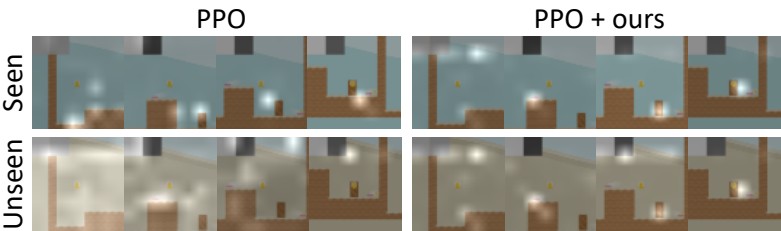

Figure 4: Visualization of activation maps via Grad-CAM in seen and unseen environments in the small-scale CoinRun. Images are aligned with similar states from various episodes for comparison.

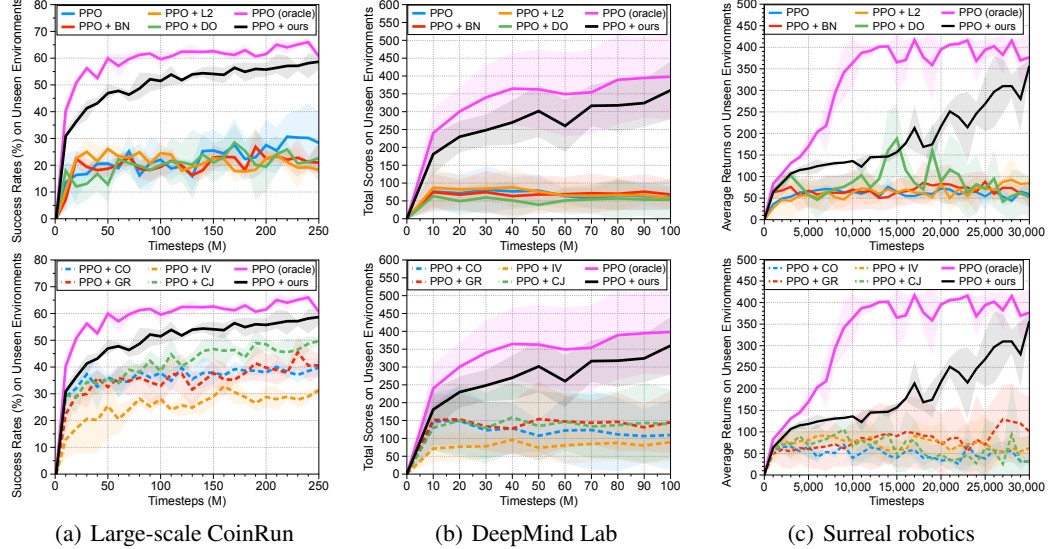

(a) Large-scale CoinRun     (b) DeepMind Lab     (c) Surreal robotics

Figure 5: The performances of trained agents in unseen environments under (a) large-scale CoinRun, (b) DeepMind Lab and (c) Surreal robotics control. The solid/dashed lines and shaded regions represent the mean and standard deviation, respectively.

**Visual interpretation**. To verify whether the trained agents can focus on meaningful and high-level information, the activation maps are visualized using Grad-CAM (Selvaraju et al., 2017) by averaging activations channel-wise in the last convolutional layer, weighted by their gradients. As shown in Figure 4, both vanilla PPO and our agents make a decision by focusing on essential objects, such as obstacles and coins in the seen environment. However, in the unseen environment, the vanilla PPO agent displays a widely distributed activation map in some cases, while our agent does not. As a quantitative metric, we measure the entropy of normalized activation maps. Specifically, we first normalize activations $\sigma_{t,h,w} \in [0,1]$, such that it represents a 2D discrete probability distribution at timestep $t$, i.e., $\sum_{h=1}^{H} \sum_{w=1}^{W} \sigma_{t,h,w} = 1$. Then, we measure the entropy averaged over the timesteps as follows: $-\frac{1}{T} \sum_{t=1}^{T} \sum_{h=1}^{H} \sum_{w=1}^{W} \sigma_{t,h,w} \log \sigma_{t,h,w}$. Note that the entropy of the activation map quantitatively measures the frequency an agent focuses on salient components in its observation. Results show that our agent produces a low entropy on both seen and unseen environments (i.e., 2.28 and 2.44 for seen and unseen, respectively), whereas the vanilla PPO agent produces a low entropy only in the seen environment (2.77 and 3.54 for seen and unseen, respectively).

**Results on large-scale experiments**. Similar to Cobbe et al. (2019), the generalization ability by training agents is evaluated on a fixed set of 500 levels of CoinRun. To explicitly separate seen and unseen environments, half of the available themes are utilized (i.e., style of backgrounds, floors, agents, and moving obstacles) for training, and the performances on 1,000 different levels consisting of unseen themes are measured.[9] As shown in Figure 5(a), our method outperforms all baseline methods by a large margin. In particular, the success rates are improved from 39.8% to 58.7% compared to the PPO with cutout (CO) augmentation proposed in Cobbe et al. (2019), showing that our agent learns generalizable representations given a limited number of seen environments.

---

[9]Oracle agents are trained on same map layouts with unseen themes to measure the optimal generalization performances on unseen visual patterns.

| | PPO | | | | PPO + ours | |
|---|---|---|---|---|---|---|
| | # of Seen Environments | Total Rewards | # of Seen Environments | Total Rewards | # of Seen Environments | Total Rewards |
| DeepMind Lab | 1 | $55.4 \pm_{33.2}$ | 16 | $218.3 \pm_{99.2}$ | 1 | $\mathbf{358.2} \pm_{81.5}$ |
| Surreal Robotics | 1 | $59.2 \pm_{31.9}$ | 25 | $168.8 \pm_{155.8}$ | 1 | $\mathbf{356.8} \pm_{15.4}$ |

Table 3: Comparison with domain randomization. The results show the mean and standard deviation averaged over three runs and the best results are indicated in bold.

### 4.3 Experiments on DeepMind Lab and Surreal robotics control

**Results on DeepMind Lab**. We also demonstrate the effectiveness of our proposed method on DeepMind Lab (Beattie et al., 2016), which is a 3D game environment in the first-person point of view with rich visual inputs. The task is designed based on the standard exploration task, where a goal object is placed in one of the rooms in a 3D maze. In this task, agents aim to collect as many goal objects as possible within 90 seconds to maximize their rewards. Once the agent collects the goal object, it receives ten points and is relocated to a random place. Similar to the small-scale CoinRun experiment, agents are trained to collect the goal object in a fixed map layout and tested in unseen environments with only changing the style of the walls and floors. We report the mean and standard deviation of the average scores across ten different map layouts, which are randomly selected. Additional details are provided in Appendix G.

Note that a simple strategy of exploring the map actively and recognizing the goal object achieves high scores because the maze size is small in this experiment. Even though the baseline agents achieve high scores by learning this simple strategy in the seen environment (see Figure 6(c) in Appendix A for learning curves), Figure 5(b) shows that they fail to adapt to the unseen environments. However, the agent trained by our proposed method achieves high scores in both seen and unseen environments. These results show that our method can learn generalizable representations from high-dimensional and complex input observations (i.e., 3D environment).

**Results on Surreal robotics control**. We evaluate our method in the Block Lifting task using the Surreal distributed RL framework (Fan et al., 2018): the Sawyer robot receives a reward if it succeeds to lift a block randomly placed on a table. We train agents on a single environment and test on five unseen environments with various styles of tables and blocks (see Appendix I for further details). Figure 5(c) shows that our method achieves a significant performance gain compared to all baselines in unseen environments while maintaining its performance in the seen environment (see Figure 13 in Appendix I), implying that our method can maintain essential properties, such as structural spatial features of the input observation.

**Comparison with domain randomization**. To further verify the effectiveness of our method, the vanilla PPO agents are trained by increasing the number of seen environments generated by randomizing rendering in a simulator, while our agent is still trained in a single environment (see Appendices G and I for further details). Table 3 shows that the performance of baseline agents can be improved with domain randomization (Tobin et al., 2017). However, our method still outperforms the baseline methods trained with more diverse environments than ours, implying that our method is more effective in learning generalizable representations than simply increasing the (finite) number of seen environments.

## 5 Conclusion

In this paper, we explore generalization in RL where the agent is required to generalize to new environments in unseen visual patterns, but semantically similar. To improve the generalization ability, we propose to randomize the first layer of CNN to perturb low-level features, e.g., various textures, colors, or shapes. Our method encourages agents to learn invariant and robust representations by producing diverse visual input observations. Such invariant features could be useful for several other related topics, like an adversarial defense in RL (see Appendix B for further discussions), sim-to-real transfer (Tobin et al., 2017; Ren et al., 2019), transfer learning (Parisotto et al., 2016; Rusu et al., 2016a;b), and online adaptation (Nagabandi et al., 2019). We provide the more detailed discussions on an extension to the dynamics generalization and failure cases of our method in Appendix J and K, respectively.

ACKNOWLEDGEMENTS

This work was supported in part by Kwanjeong Educational Foundation Scholarship and Sloan Research Fellowship. We also thank Sungsoo Ahn, Jongwook Choi, Wilka Carvalho, Yijie Guo, Yunseok Jang, Lajanugen Logeswaran, Sejun Park, Sungryull Sohn, Ruben Villegas, and Xinchen Yan for helpful discussions.

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

# Appendix: Network Randomization: A Simple Technique for Generalization in Deep Reinforcement Learning

## A  LEARNING CURVES

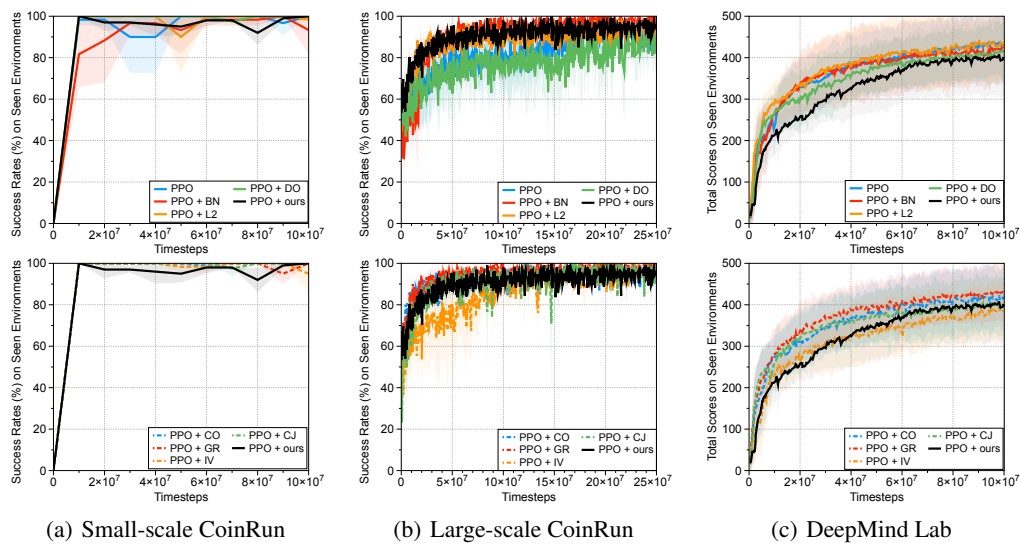

(a) Small-scale CoinRun     (b) Large-scale CoinRun     (c) DeepMind Lab

Figure 6: Learning curves on (a) small-scale, (b) large-scale CoinRun and (c) DeepMind Lab. The solid line and shaded regions represent the mean and standard deviation, respectively, across three runs.

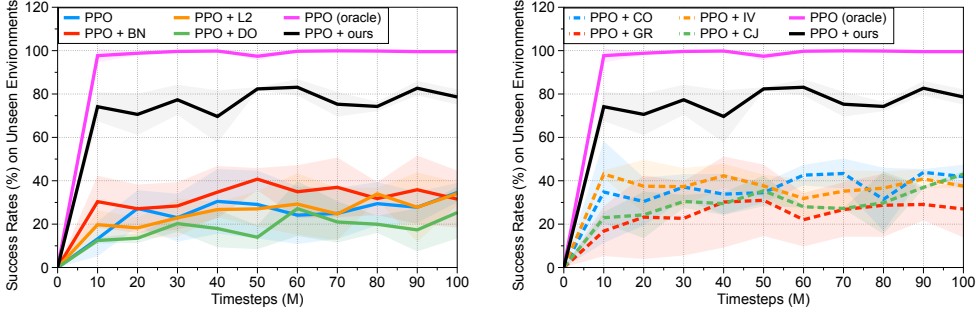

(a) Comparison with regularization techniques     (b) Comparison with data augmentation techniques

Figure 7: The performance in unseen environments in small-scale CoinRun. The solid/dashed line and shaded regions represent the mean and standard deviation, respectively, across three runs.

## B  ROBUSTNESS AGAINST ADVERSARIAL ATTACKS

The adversarial (visually imperceptible) perturbation (Szegedy et al., 2014) to clean input observations can induce the DNN-based policies to generate an incorrect decision at test time (Huang et al., 2017; Lin et al., 2017). This undesirable property of DNNs has raised major security concerns. In this section, we evaluate if the proposed method can improve the robustness on adversarial attacks. Our method is expected to improve the robustness against such adversarial attacks because the agents are trained with randomly perturbed inputs. To verify that the proposed method can improve the robustness to adversarial attacks, the adversarial samples are generated using FGSM (Goodfellow et al., 2015) by perturbing inputs to the opposite direction to the most probable action initially predicted by the policy:

$$s_{\text{adv}} = s - \varepsilon \text{sign} \left( \nabla_s \log \pi(a^* | s; \theta) \right),$$

where $\varepsilon$ is the magnitude of noise and $a^* = \arg\max_a \pi(a|s;\theta)$ is the action from the policy. Table 4 shows that our proposed method can improve the robustness against FGSM attacks with $\varepsilon = 0.01$, which implies that hidden representations of trained agents are more robust.

| | Small-Scale CoinRun | | Large-Scale CoinRun | | DeepMind Lab | |
|---|---|---|---|---|---|---|
| | Clean | FGSM | Clean | FGSM | Clean | FGSM |
| PPO | 100 | 61.5 (-38.5) | 96.2 | 77.4 (-19.5) | 352.5 | 163.5 (-53.6) |
| PPO + ours | 100 | 88.0 **(-12.0)** | 99.6 | 84.4 **(-15.3)** | 368.0 | 184.0 **(-50.0)** |

Table 4: Robustness against FGSM attacks on training environments. The values in parentheses represent the relative reductions from the clean samples.

## C  DETAILS FOR TRAINING AGENTS USING PPO

**Policy optimization**. For all baselines and our methods, PPO is utilized to train the policies. Specifically, we use a discount factor $\gamma = 0.999$, a generalized advantage estimator (GAE) Schulman et al. (2016) parameter $\lambda = 0.95$, and an entropy bonus (Williams & Peng, 1991) of 0.01 to ensure sufficient exploration. We extract 256 timesteps per rollout, and then train the agent for 3 epochs with 8 mini-batches. The Adam optimizer (Kingma & Ba, 2015) is used with the starting learning rate 0.0005. We run 32 environments simultaneously during training. As suggested in Cobbe et al. (2019), two boxes are painted in the upper-left corner, where their color represents the $x$- and $y$-axis velocity to help the agents quickly learn to act optimally. In this way, the agent does not need to memorize previous states, so a simple CNN-based policy without LSTM can effectively perform in our experimental settings.

**Data augmentation methods**. In this paper, we compare a variant of cutout (DeVries & Taylor, 2017) proposed in Cobbe et al. (2019), grayout, inversion, and color jitter (Cubuk et al., 2019). Specifically, the cutout augmentation applies a random number of boxes in random size and color to the input, the grayout method averages all three channels of the input, the inversion method inverts pixel values by a 50% chance, and the color jitter changes the characteristics of images commonly used for data augmentation in computer vision tasks: brightness, contrast, and saturation. For every timestep in the cutout augmentation, we first randomly choose the number of boxes from zero to five, assign them a random color and size, and place them in the observation. For the color jitter, the parameters for brightness, contrast, and saturation are randomly chosen in [0.5,1.5].[10] For each episode, the parameters of these methods are randomized and fixed such that the same image pre-processing is applied within an episode.

## D  DIFFERENT TYPES OF RANDOM NETWORKS

In this section, we apply random networks to various locations in the network architecture (see Figure 9) and measure the performance in large-scale CoinRun without the feature matching loss. For all methods, a single-layer CNN is used with a kernel size of 3, and its output tensor is padded in order to be in the same dimension as the input tensor. As shown in Figure 8, the performance of unseen environments decreases as the random network is placed in higher layers. On the other hand, the random network in residual connections improves the generalization performance, but it does not outperform the case when a random network is placed at the beginning of the network, meaning that randomizing only the local features of inputs can be effective for a better generalization performance.

---

[10]For additional details, see `https://pytorch.org/docs/stable/_modules/torchvision/transforms/transforms.html#ColorJitter`.

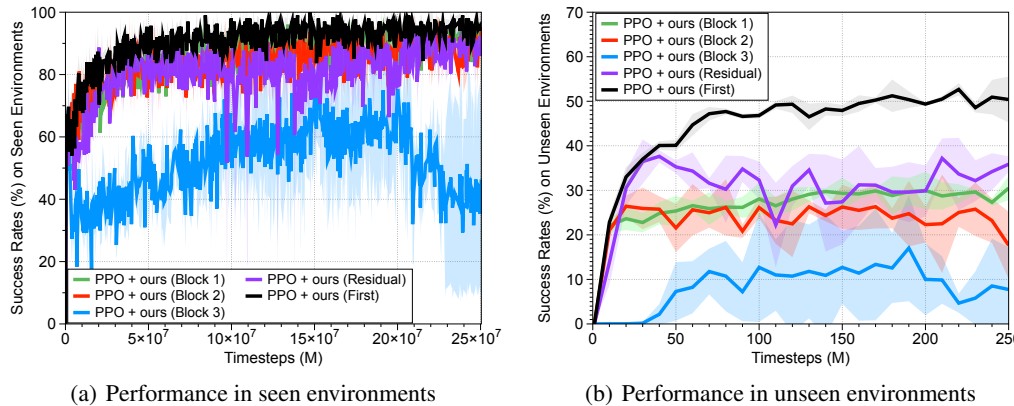

(a) Performance in seen environments

(b) Performance in unseen environments

Figure 8: The performance of random networks in various locations in the network architecture on (a) seen and (b) unseen environments in large-scale CoinRun. We show the mean performances averaged over three different runs, and shaded regions represent the standard deviation.

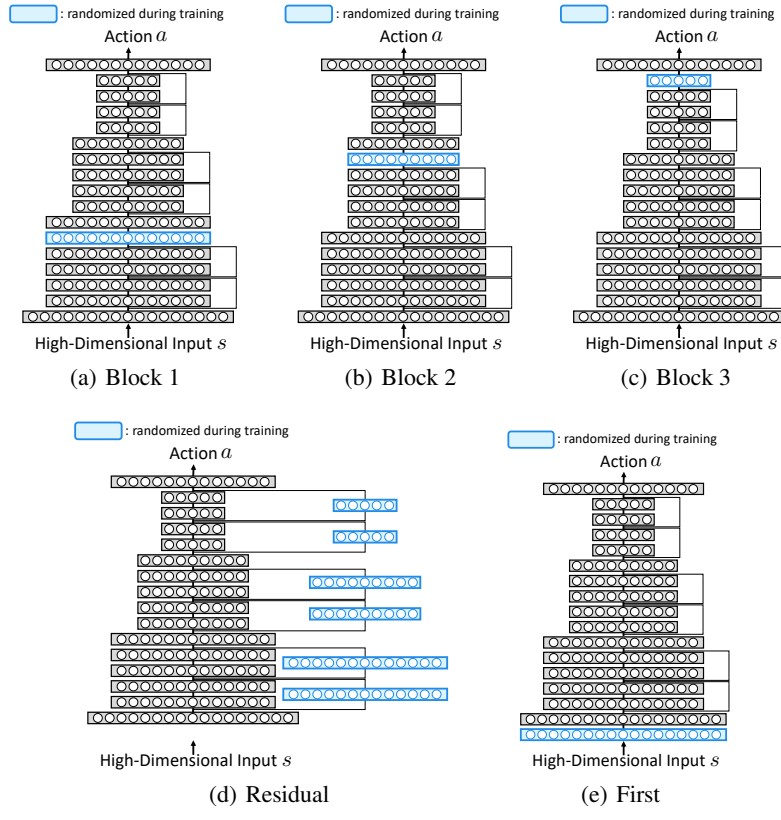

Figure 9: Network architectures with random networks in various locations. Only convolutional layers and the last fully connected layer are displayed for conciseness.

## E   ENVIRONMENTS IN SMALL-SCALE COINRUN

For small-scale CoinRun environments, we consider a fixed map layout with two moving obstacles and measure the performance of the trained agents by changing the style of the backgrounds (see Figure 10). Below is the list of seen and unseen backgrounds in this experiment:

○ Seen background:

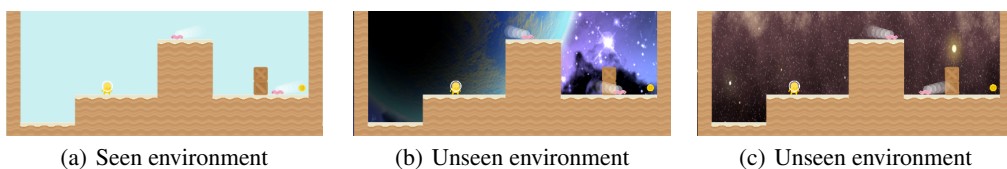

(a) Seen environment  (b) Unseen environment  (c) Unseen environment

Figure 10: Examples of seen and unseen environments in small-scale CoinRun.

- `kenney/Backgrounds/blue_desert.png`
○ Unseen backgrounds:
  - `kenney/Backgrounds/colored_desert.png`
  - `kenney/Backgrounds/colored_grass.png`
  - `backgrounds/game-backgrounds/seabed.png`
  - `backgrounds/game-backgrounds/G049_OT000_002A__background.png`
  - `backgrounds/game-backgrounds/Background.png`
  - `backgrounds/game-backgrounds/Background (4).png`
  - `backgrounds/game-backgrounds/BG_only.png`
  - `backgrounds/game-backgrounds/bg1.png`
  - `backgrounds/game-backgrounds/G154_OT000_002A__background.png`
  - `backgrounds/game-backgrounds/Background (5).png`
  - `backgrounds/game-backgrounds/Background (2).png`
  - `backgrounds/game-backgrounds/Background (3).png`
  - `backgrounds/background-from-glitch-assets/background.png`
  - `backgrounds/spacebackgrounds-0/deep_space_01.png`
  - `backgrounds/spacebackgrounds-0/spacegen_01.png`
  - `backgrounds/spacebackgrounds-0/milky_way_01.png`
  - `backgrounds/spacebackgrounds-0/deep_sky_01.png`
  - `backgrounds/spacebackgrounds-0/space_nebula_01.png`
  - `backgrounds/space-backgrounds-3/Background-1.png`
  - `backgrounds/space-backgrounds-3/Background-2.png`
  - `backgrounds/space-backgrounds-3/Background-3.png`
  - `backgrounds/space-backgrounds-3/Background-4.png`
  - `backgrounds/background-2/airadventurelevel1.png`
  - `backgrounds/background-2/airadventurelevel2.png`
  - `backgrounds/background-2/airadventurelevel3.png`
  - `backgrounds/background-2/airadventurelevel4.png`

## F   ENVIRONMENTS IN LARGE-SCALE COINRUN

In CoinRun, there are 34 themes for backgrounds, 6 for grounds, 5 for agents, and 9 for obstacles. For the large-scale CoinRun experiment, we train agents on a fixed set of 500 levels of CoinRun using half of the available themes and measure the performances on 1,000 different levels consisting of unseen themes. Specifically, the following is a list of seen and unseen themes used in this experiment:

○ Seen backgrounds:
  - `kenney/Backgrounds/blue_desert.png`
  - `kenney/Backgrounds/blue_grass.png`
  - `kenney/Backgrounds/blue_land.png`
  - `kenney/Backgrounds/blue_shroom.png`

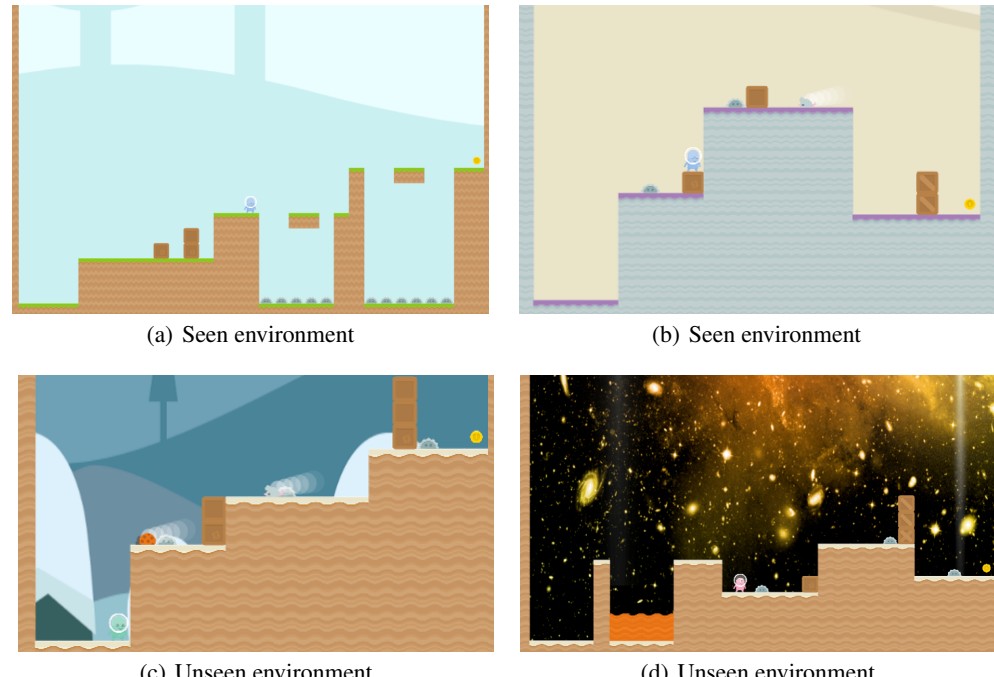

(a) Seen environment                    (b) Seen environment

(c) Unseen environment                  (d) Unseen environment

Figure 11: Examples of seen and unseen environments in large-scale CoinRun.

- `kenney/Backgrounds/colored_desert.png`
- `kenney/Backgrounds/colored_grass.png`
- `kenney/Backgrounds/colored_land.png`
- `backgrounds/game-backgrounds/seabed.png`
- `backgrounds/game-backgrounds/G049_OT000_002A__background.png`
- `backgrounds/game-backgrounds/Background.png`
- `backgrounds/game-backgrounds/Background (4).png`
- `backgrounds/game-backgrounds/BG_only.png`
- `backgrounds/game-backgrounds/bg1.png`
- `backgrounds/game-backgrounds/G154_OT000_002A__background.png`
- `backgrounds/game-backgrounds/Background (5).png`
- `backgrounds/game-backgrounds/Background (2).png`
- `backgrounds/game-backgrounds/Background (3).png`

○ Unseen backgrounds:

- `backgrounds/background-from-glitch-assets/background.png`
- `backgrounds/spacebackgrounds-0/deep_space_01.png`
- `backgrounds/spacebackgrounds-0/spacegen_01.png`
- `backgrounds/spacebackgrounds-0/milky_way_01.png`
- `backgrounds/spacebackgrounds-0/ez_space_lite_01.png`
- `backgrounds/spacebackgrounds-0/meyespace_v1_01.png`
- `backgrounds/spacebackgrounds-0/eye_nebula_01.png`
- `backgrounds/spacebackgrounds-0/deep_sky_01.png`
- `backgrounds/spacebackgrounds-0/space_nebula_01.png`
- `backgrounds/space-backgrounds-3/Background-1.png`
- `backgrounds/space-backgrounds-3/Background-2.png`
- `backgrounds/space-backgrounds-3/Background-3.png`

- `backgrounds/space-backgrounds-3/Background-4.png`
- `backgrounds/background-2/airadventurelevel1.png`
- `backgrounds/background-2/airadventurelevel2.png`
- `backgrounds/background-2/airadventurelevel3.png`
- `backgrounds/background-2/airadventurelevel4.png`

○ Seen grounds:
  - `Dirt`
  - `Grass`
  - `Planet`
○ Unseen grounds:
  - `Sand`
  - `Snow`
  - `Stone`

○ Seen player themes:
  - `Beige`
  - `Blue`
○ Unseen player themes:
  - `Green`
  - `Pink`
  - `Yellow`

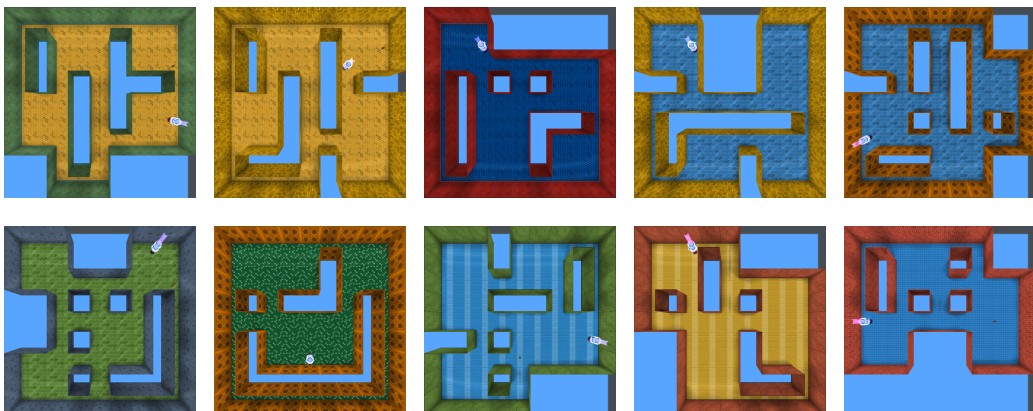

Figure 12: The top-down view of the trained map layouts.

## G  ENVIRONMENTS ON DEEPMIND LAB

**Dataset**. Among the styles (textures and colors) provided for the 3D maze in the DeepMind Lab, we take ten different styles of floors and walls, respectively (see the list below). We construct a training dataset by randomly choosing a map layout and assigning a theme among ten floors and walls, respectively. The domain randomization method compared in Table 3 uses four floors and four wall themes (16 combinations in total). Trained themes are randomly chosen before training and their combinations are considered to be seen environments. To evaluate the generalization ability, we measure the performance of trained agents on unseen environments by changing the styles of walls and floors. Domain randomization has more seen themes than the other methods, so all methods are compared with six floors and six walls (36 combinations in total), which are unseen for all methods. The mean and standard deviation of the average scores across ten different map layouts are reported in Figure 12.

○ Floor themes:
- `lg_style_01_floor_orange`
- `lg_style_01_floor_blue`
- `lg_style_02_floor_blue`
- `lg_style_02_floor_green`
- `lg_style_03_floor_green`
- `lg_style_03_floor_blue`
- `lg_style_04_floor_blue`
- `lg_style_04_floor_orange`
- `lg_style_05_floor_blue`
- `lg_style_05_floor_orange`

○ Wall themes:
- `lg_style_01_wall_green`
- `lg_style_01_wall_red`
- `lg_style_02_wall_yellow`
- `lg_style_02_wall_blue`
- `lg_style_03_wall_orange`
- `lg_style_03_wall_gray`
- `lg_style_04_wall_green`
- `lg_style_04_wall_red`
- `lg_style_05_wall_red`
- `lg_style_05_wall_yellow`

**Action space**. Similar to IMPALA (Espeholt et al., 2018), the agent can take eight actions from the DeepMind Lab native action samples: {Forward, Backward, Move Left, Move Right, Look Left, Look Right, Forward + Look Left, and Forward + Look Right}. Table 5 describes the detailed mapping.

| Action | DeepMind Lab Native Action |
|---|---|
| Forward | `[  0,  0,   0,   1,  0,  0,  0]` |
| Backward | `[  0,  0,   0,  -1,  0,  0,  0]` |
| Move Left | `[  0,  0,  -1,   0,  0,  0,  0]` |
| Move Right | `[  0,  0,   1,   0,  0,  0,  0]` |
| Look Left | `[-20,  0,   0,   0,  0,  0,  0]` |
| Look Right | `[ 20,  0,   0,   0,  0,  0,  0]` |
| Forward + Look Left | `[-20,  0,   0,   1,  0,  0,  0]` |
| Forward + Look Right | `[ 20,  0,   0,   1,  0,  0,  0]` |

Table 5: Action set used in the DeepMind Lab experiment. The DeepMind Lab native action set consists of seven discrete actions encoded in integers ([L,U] indicates the lower/upper bound of the possible values): 1) yaw (left/right) rotation by pixel [-512,512], 2) pitch (up/down) rotation by pixel [-512,512], 3) horizontal move [-1,1], 4) vertical move [-1,1], 5) fire [0,1], 6) jump [0,1], and 7) crouch [0,1].

# H  EXPERIMENTS ON DOGS AND CATS DATABASE

**Dataset**. The original database is a set of 25,000 images of dogs and cats for training and 12,500 images for testing. Similar to Kim et al. (2019), the data is manually categorized according to the color of the animal: bright or dark. Biased datasets are constructed such that the training set consists of bright dogs and dark cats, while the test and validation sets contain dark dogs and bright cats. Specifically, training, validation, and test sets consist of 10,047, 1,000, and 5,738 images, respectively.[11] ResNet-18 (He et al., 2016) is trained with an initial learning rate chosen from {0.05, 0.1}

---

[11]The code is available at: https://github.com/feidfoe/learning-not-to-learn/tree/master/dataset/dogs_and_cats.

and then dropped by 0.1 at 50 epochs with a total of 100 epochs. We use the Nesterov momentum of 0.9 for SGD, a mini-batch size chosen from {32, 64}, and the weight decay set to 0.0001. We report the training and test set accuracies with the hyperparameters chosen by validation. Unlike Kim et al. (2019), we do not use ResNet-18 pre-trained with ImageNet (Russakovsky et al., 2015) in order to avoid inductive bias from the pre-trained dataset.

## I  EXPERIMENTAL RESULTS ON SURREAL ROBOT MANIPULATION

Our method is evaluated in the Block Lifting task using the Surreal distributed RL framework (Fan et al., 2018). In this task, the Sawyer robot receives a reward if it successfully lifts a block randomly placed on a table. Following the experimental setups in (Fan et al., 2018), the hybrid CNN-LSTM architecture (see Figure 13(a)) is chosen as the policy network and a distributed version of PPO (i.e., actors collect massive amount of trajectories and the centralized learner updates the model parameters using PPO) is used to train the agents.[12] Agents take $84 \times 84$ observation frames with proprioceptive features (e.g., robot joint positions and velocities) and output the mean and log of the standard deviation for each action dimension. The actions are then sampled from the Gaussian distribution parameterized by the output. Agents are trained on a single environment and tested on five unseen environments with various styles of table, floor, and block, as shown in Figure 14. For the Surreal robot manipulation experiment, the vanilla PPO agent is trained on 25 environments generated by changing the styles of tables and boxes. Specifically, we use {blue, gray, orange, white, purple} and {red, blue, green, yellow, cyan} for table and box, respectively.

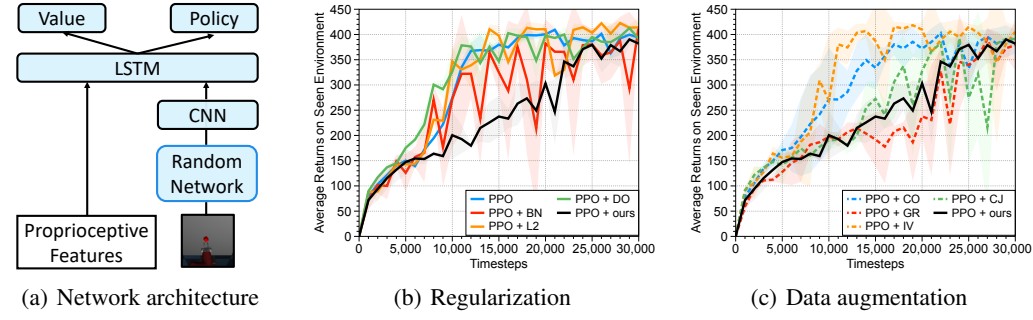

|  (a) Network architecture  |  (b) Regularization  |  (c) Data augmentation  |

Figure 13: (a) An illustration of network architectures for the Surreal robotics control experiment, and learning curves with (b) regularization and (c) data augmentation techniques. The solid line and shaded regions represent the mean and standard deviation, respectively, across three runs.

## J  EXTENSION TO DOMAINS WITH DIFFERENT DYNAMICS

In this section, we consider an extension to the generalization on domains with different dynamics. Similar to dynamics randomization (Peng et al., 2018), one can expect that our idea can be useful for improving the dynamics generalization. To verify this, we conduct an experiment on CartPole and Hopper environments where an agent takes proprioceptive features (e.g., positions and velocities). The goal of CartPole is to prevent the pole from falling over, while that of Hopper is to make an one-legged robot hop forward as fast as possible, respectively. Similar to the randomization method we applied to visual inputs, we introduce a random layer between the input and the model. As a natural extension of the proposed method, we consider performing the convolution operation by multiplying a $d \times d$ diagonal matrix to $d$-dimensional input states. For every training iteration, the elements of the matrix are sampled from the standard uniform distribution $U(0.8, 1.2)$. One can note that this method can randomize the amplitude of input states while maintaining the intrinsic information (e.g., sign of inputs). Following Packer et al. (2018); Zhou et al. (2019), we measure the performance of the trained agents on unseen environments with a different set of dynamics parameters, such as mass, length, and force. Specifically, for CartPole experiments, similar to Packer et al. (2018), the policy

---

[12]We used a reference implementation with two actors in: https://github.com/SurrealAI/surreal.

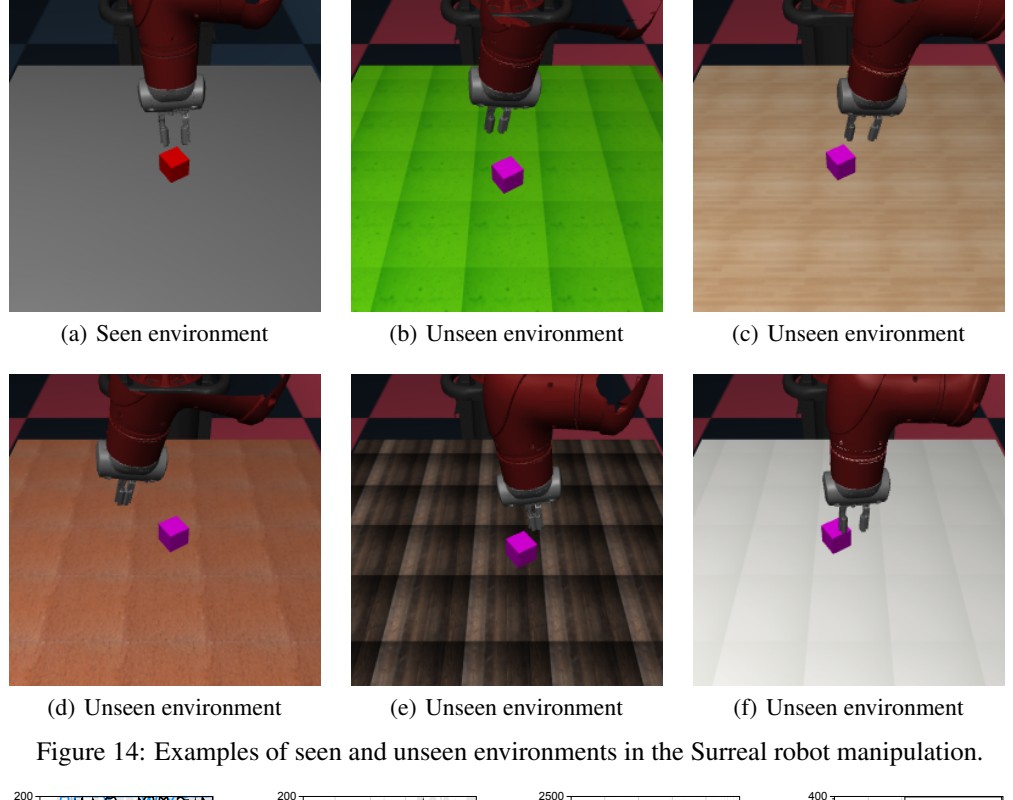

(a) Seen environment      (b) Unseen environment      (c) Unseen environment

(d) Unseen environment      (e) Unseen environment      (f) Unseen environment

Figure 14: Examples of seen and unseen environments in the Surreal robot manipulation.

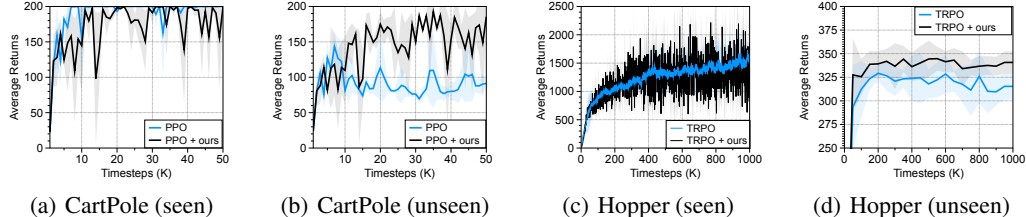

(a) CartPole (seen)     (b) CartPole (unseen)     (c) Hopper (seen)     (d) Hopper (unseen)

Figure 15: Performances of trained agents in seen and unseen environments under (a/b) CartPole and (c/d) Hopper. The solid/dashed lines and shaded regions represent the mean and standard deviation, respectively.

and value functions are multi-layer perceptrons (MLPs) with two hidden layers of 64 units each and hyperbolic tangent activation and the Proximal Policy Optimization (PPO) (Schulman et al., 2017) method is used to train the agents. The parameters of the training environment are fixed at the default values in the implementations from Gym, while force, length, and mass of environments are sampled from $[1, 5] \cup [15, 20]$, $[0.05, 0.25] \cup [0.75, 1.0]$, $[0.01, 0.05] \cup [0.5, 1.0]$ that the policy has never seen in any stage of training.[13] For Hopper experiments, similar to Zhou et al. (2019), the policy is a MLP with two hidden layers of 32 units each and ReLU activation and value function is a linear model. The trust region policy optimization (TRPO) (Schulman et al., 2015) method is used to train the agents. The mass of the training environment is sampled from $\{1.0, 2.0, 3.0, 4.0, 5.0\}$, while it is sampled from $\{6.0, 7.0, 8.0\}$ during testing.[14] Figure 15 reports the mean and standard deviation across 3 runs. Our simple randomization improves the performance of the agents in unseen environments, while achieving performance comparable to seen environments. We believe that this evidences a wide applicability of our idea beyond visual changes.

---

[13]We used a reference implementation in https://bair.berkeley.edu/blog/2019/03/18/rl-generalization.

[14]We used a reference implementation with two actors in: https://github.com/Wenxuan-Zhou/EPI.

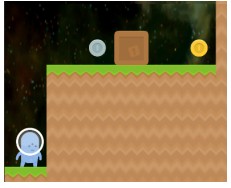 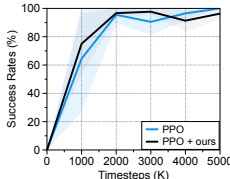 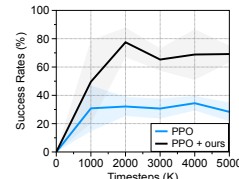 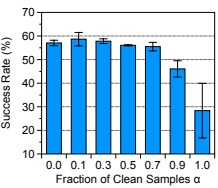

(a) CoinRun with good and bad coins  (b) Performance in seen environments  (c) Performance in unseen environments  (d) Performance in unseen environments

Figure 16: (a) Modified CoinRun with good and bad coins. The performances on (b) seen and (c) unseen environments. The solid line and shaded regions represent the mean and standard deviation, respectively, across three runs. (d) Average success rates on large-scale CoinRun for varying the fraction of clean samples during training. Noe that $\alpha = 1$ corresponds to vanilla PPO agents.

## K  FAILURE CASE OF OUR METHODS

In this section, we verify whether the proposed method can handle color (or texture)-conditioned RL tasks. One might expect that such RL tasks can be difficult for our methods to work because of the randomization. For example, our methods would fail if we consider an extreme seek-avoid object gathering setup, where the agent must learn to collect good objects and avoid bad objects which have the same shape but different color. However, we remark that our method would not always fail for such tasks if other environmental factors (e.g., the shape of objects in Collect Good Objects in DeepMind Lab (Beattie et al., 2016)) are available to distinguish them. To verify this, we consider a modified CoinRun environment where the agent must learn to collect good objects (e.g., gold coin) and avoid bad objects (e.g., silver coin). Similar to the small-scale CoinRun experiment, agents are trained to collect the goal object in a fixed map layout (see Figure 16(a)) and tested in unseen environments with only changing the style of the background. Figure 16(b) shows that our method can work well for such color-conditioned RL tasks because a trained agent can capture the other factors such as a location to perform this task. Besides, our method achieves a significant performance gain compared to vanilla PPO agent in unseen environments as shown in Figure 16(c).

As another example, in color-matching tasks such as the keys doors puzzle in DeepMind Lab (Beattie et al., 2016), the agent must collect colored keys to open matching doors. Even though this task is color-conditioned, a policy trained with our method can perform well because the same colored objects will have the same color value even after randomization, i.e., our randomization method still maintains the structure of input observation. This evidences the wide applicability of our idea. We also remark that our method can handle more extreme corner cases by adjusting the fraction of clean samples during training. In summary, we believe that the proposed method covers a broad scope of generalization across low-level transformations in the observation space features.

## L  ABLATION STUDY FOR FRACTION OF CLEAN SAMPLES

We investigate the effect of the fraction of clean samples. Figure 16(d) shows that the best unseen performance is achieved when the fraction of clean samples is 0.1 on large-scale CoinRun.

## M  TRAINING ALGORITHM

---
**Algorithm 1** PPO + random networks, Actor-Critic Style

---
    **for** iteration= $1, 2, \cdots$ **do**
        Sample the parameter $\phi$ of random networks from prior distribution $P(\phi)$
        **for** actor= $1, 2, \cdots, N$ **do**
            Run policy $\pi(a|f(s;\phi);\theta)$ in the given environment for $T$ timesteps
            Compute advantage estimates
        **end for**
        Optimize $\mathcal{L}^{\texttt{random}}$ in equation (3) with respect to $\theta$
    **end for**

---

# N VISUALIZATION OF HIDDEN FEATURES

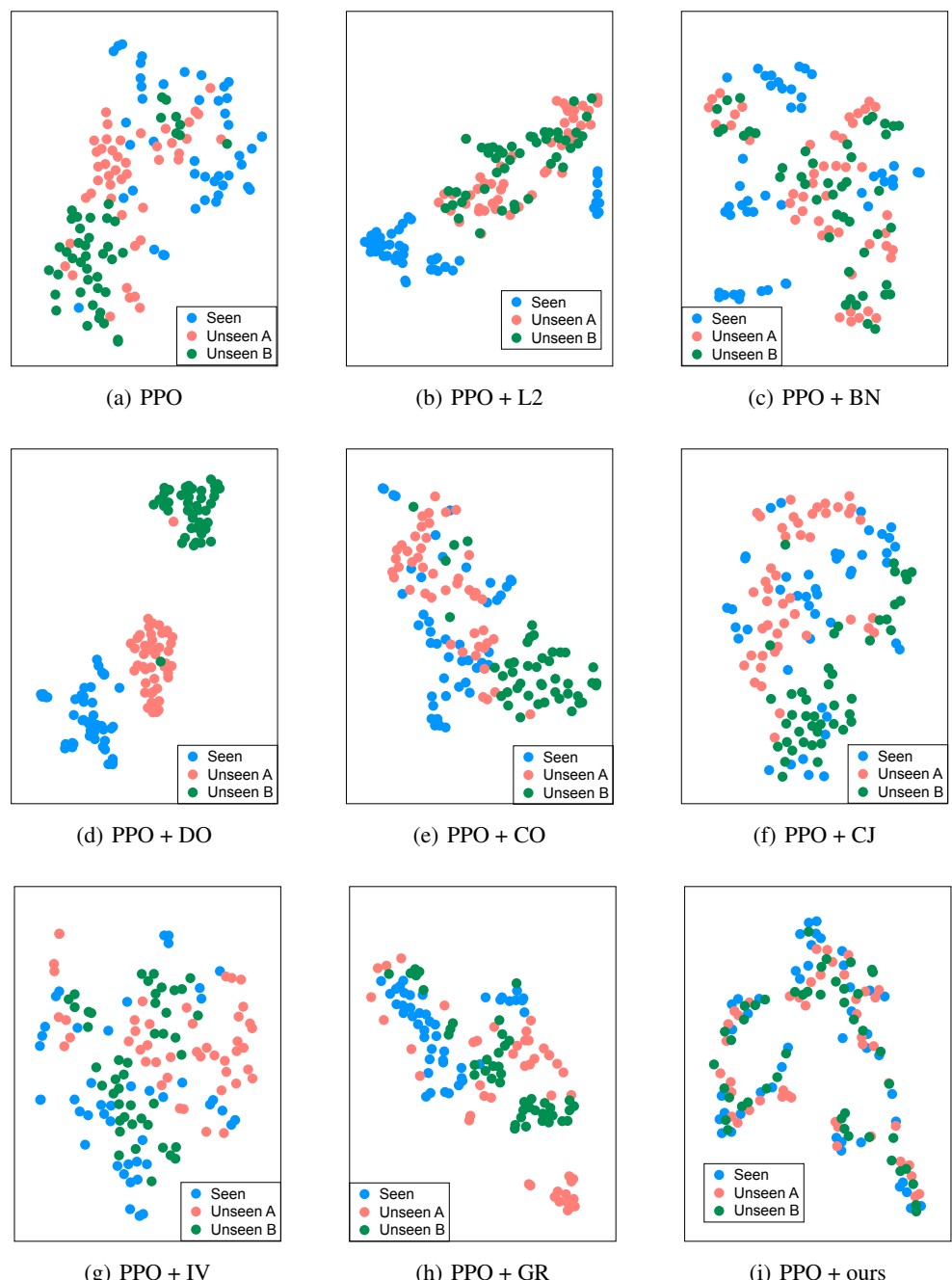

Figure 17: Visualization of the hidden representation of trained agents optimized by (a) PPO, (b) PPO + L2, (c) PPO + BN, (d) PPO + DO, (e) PPO + CO, (f) PPO + GR, (g) PPO + IV, (h) PPO + CJ, and (I) PPO + ours using t-SNE. The point colors indicate the environments of the corresponding observations.

