# OpenReview forum: "Network Randomization: A Simple Technique for Generalization in Deep Reinforcement Learning"
_ICLR.cc/2020/Conference — Accept (Poster)_

### Official Review · AnonReviewer3 · 2019-10-23
**Official Blind Review #3**

**Rating:** 6

**Review:**

This paper proposes methods to improve generalization in deep reinforcement learning with an emphasis on unseen environments. The main contribution is essentially a data augmentation technique that perturbs the input observations using a noise generated from the range space of a random convolutional network. The empirical results look impressive and demonstrate the effectiveness of the method. The experiments are thorough (includes even adversarial attack) and the core method is novel as far as I am aware.

That said, I have a couple of concerns regarding this paper and I would be willing to change my score if authors can address these.

1) Feature matching loss (Eq 2) is presented as a novel contribution without referring to related work in semisupervised learning literature. This is essentially consistency training. See:
a) Miyato, Takeru, et al. "Virtual adversarial training: a regularization method for supervised and semi-supervised learning." IEEE transactions on pattern analysis and machine intelligence 41.8 (2018): 1979-1993.
b) Xie, Qizhe, et al. "Unsupervised data augmentation." arXiv preprint arXiv:1904.12848 (2019).

2) The main contribution appears to be a data augmentation technique where we add a random neural net based perturbation to the state. My question is:

*Why don't you first evaluate this on computer vision tasks given that the core idea is data augmentation for images?*

If this technique is so powerful, shouldn't this do a great job in CIFAR10, Imagenet etc? Instead authors only provide a niche example (bright vs dark cat/dogs).

If this can compete with top augmentation techniques on Imagenet (e.g. autoagument), then it can explain the RL performance. Otherwise, please provide some intuition on why this works so well on RL but not as well on computer vision tasks. Is it the unseen environment diversity of RL challenges?

3) While proposed method performs well on the benchmarks, it is not clear whether authors compare to the state-of-the-art algorithms. For each task (CoinRun, DeepMind Lab, etc), please explicitly state the best prior result (e.g. Espeholt et al, Tobin et al, Cobbe et al etc) so that proposed method's performance can be better assessed.

-------------------------

After rebuttal: Authors addressed most of my comments. I also found the new experimental results (Fig 5 and 7) very insightful. I increase my score to Weak Accept.

For future improvement: More realistic experiments on computer vision tasks (besides cats and dogs) would be welcome. Otherwise, please justify why proposed strategy is particularly good for RL (rather than traditional computer vision benchmarks) in boosting robustness to new domains.

**Experience Assessment:**

I have read many papers in this area.

**Review Assessment: Checking Correctness Of Derivations And Theory:**

N/A

**Review Assessment: Checking Correctness Of Experiments:**

I assessed the sensibility of the experiments.

**Review Assessment: Thoroughness In Paper Reading:**

I read the paper at least twice and used my best judgement in assessing the paper.

---

> ### Author Response · Authors · 2019-11-15
> **Response to R3**
>
> We appreciate your valuable comments, efforts and times on our paper. As you and R4 mentioned, we introduce a simple, yet powerful solution for generalization in deep RL, and show that it achieves significant performance gains in comparison to many known regularization and data augmentation techniques in various environments. Our responses to all your questions are provided below. Revised parts in the new draft are colored by red (in particular, we updated Section 3, 4 and 5, Appendix A, L, M, and N, and Figure 5, 7, 16, and 17).
>
> Q1. Related work
>
> A1. We cited relevant works that leveraged feature matching loss in Section 3.1, including [1, 2] in semi-supervised learning. Thank you very much for your suggestion.
>
> Q2. Random networks on computer vision tasks
>
> A2. Our method is designed for improving generalization across different data distributions (between training and testing). Namely, the purpose of our method is different from conventional data augmentation methods. Hence, its effect may not be significant for standard computer vision tasks (e.g., image classification tasks on CIFAR-10 and ImageNet) which consider the generalization in the same data distribution. We remark that "bright vs dark cat/dogs" resembles our problem setting in that the training and test datasets are sampled from a different distribution, so this is the reason why we put this experiment in our paper. Nevertheless, an extension of our method to other computer vision tasks should be an interesting future direction to explore.
>
> Q3. State-of-the-art methods
>
> A3. In our setting where a domain-specific simulator with various control parameters is not available, the cutout-variant method proposed in [3] (published in ICML 2019) is the state-of-the-art method (denoted as PPO+CO in our paper) and our method outperforms it by a large margin (see Figure 5). In addition, domain randomization [4] that can also be regarded as another state-of-the-art method when the domain-specific simulator is available. Our method outperforms it even without utilizing the simulator (see Table 3). We think this is strong evidence that the proposed method is significantly more effective than any prior method. Furthermore, following R4's suggestion, we included the performance of an agent trained directly on unseen environments to Figure 5 and 7 of the revised draft, which shows an upper bound performance of any method for our purpose. Here, our method is shown to perform very well quite close to this upper bound performance.
>
> [1] Miyato, T., Maeda, S.I., Koyama, M. and Ishii, S., Virtual adversarial training: a regularization method for supervised and semi-supervised learning. IEEE transactions on pattern analysis and machine intelligence, 41(8), pp.1979-1993, 2018.
>
> [2] Xie, Q., Dai, Z., Hovy, E., Luong, M.T. and Le, Q.V., Unsupervised data augmentation. arXiv preprint arXiv:1904.12848, 2019.
>
> [3] Karl Cobbe, Oleg Klimov, Chris Hesse, Taehoon Kim, and John Schulman. Quantifying generalization in reinforcement learning. In ICML, 2019.
>
> [4] Josh Tobin, Rachel Fong, Alex Ray, Jonas Schneider, Wojciech Zaremba, Pieter Abbeel. Domain Randomization for Transferring Deep Neural Networks from Simulation to the Real World. In IROS, 2017.

---

### Official Review · AnonReviewer5 · 2019-10-28
**Official Blind Review #5**

**Rating:** 3

**Review:**

This work proposes using a randomly parameterized convolutional layer as additional processing of the input observation to provide data augmentation to make policies more robust to environments with different observation spaces. The empirical results are thorough, comparing with other regularization techniques, including dropout, L2 regularization, and batch normalization with the same policy gradient method, PPO on a variety of generalization in RL benchmarks. There are additional experiments of this method to check that it actually removes visual bias in a computer vision problem better than other methods.

They also incorporate a feature matching loss that explicitly forces the learned representations of equivalent states to be close in L2 distance. While the empirical results are impressive, it is hard to feel excited about this work, which relies on the inductive bias of a randomly parameterized convolution layer to modify the texture of the observation and show it works in certain settings. I'd like more discussion and showcasing of failure modes, it seems that this wouldn't work for settings where the train and test environments are different in ways beyond texture and changes in small objects, and additional analysis in terms of the dogs and cats database about why it performs so much better than other methods. What exactly is the desired and meaningful information in images that a random convolution layer can keep while removing something that is able to generalize to different shades of cats and dogs? Why would this perform better than grayscaling?  What about grayscaling and additive Gaussian noise?

The comparison of PPO is also unfair in that the author's method uses an ensemble of policies to act, which other methods do not. A more fair comparison would use ensembles in all other baselines as well or results showing how their method performs without this ensemble.

Overall, the presentation, analysis, and writing can all be improved to match the strong empirical results produced.

**Experience Assessment:**

I have read many papers in this area.

**Review Assessment: Checking Correctness Of Derivations And Theory:**

N/A

**Review Assessment: Checking Correctness Of Experiments:**

I assessed the sensibility of the experiments.

**Review Assessment: Thoroughness In Paper Reading:**

N/A

---

> ### Author Response · Authors · 2019-11-15
> **Response to R5**
>
> We appreciate your valuable comments, efforts and times on our paper. As R3 and R4 mentioned, we introduce a simple, yet powerful solution for generalization in deep RL, and show that it achieves significant performance gains in comparison to many known regularization and data augmentation techniques in various environments. Our responses to all your questions are provided below. Revised parts in the new draft are colored by red (in particular, we updated Section 3, 4 and 5, Appendix A, L, M, and N, and Figure 5, 7, 16, and 17).
>
> Q1. Potential failure case and beyond visual changes
>
> A1. Thank you for the suggestion. As R4 also pointed out, some color (or texture)-conditioned RL tasks can be difficult for our methods to work. However, we remark that our method would not always fail for such tasks (please see our response A1 for R4, Section 5 and Appendix M of the revised draft). Furthermore, to address your concern about limitations, we evaluate our randomization idea on domains with different dynamics. Specifically, we conduct an experiment on the CartPole and Hopper where an agent takes proprioceptive features (e.g., positions and velocities) by changing its dynamics (see Appendix L of the revised draft for more details). Similar to the randomization method we applied to visual inputs, we introduce a random layer between the input and the model. As a natural extension of the proposed method, we consider performing the convolution operation by multiplying a $d \times d$ diagonal matrix to $d$-dimensional input states. For every training iteration, the elements of the matrix are sampled from the standard uniform distribution $U(0.8,1.2)$. One can note that this method can randomize the amplitude of input states while maintaining the intrinsic information (e.g., sign of inputs). Following [1, 2], we measure the performance of the trained agents on unseen environments with a different set of dynamics parameters such as mass. Our simple randomization improves the performance of the agents in unseen environments while achieving performance comparable to seen environments. We believe that this provides evidence for the wide applicability of our high-level idea of using randomization for achieving better generalization beyond visual changes. We added related results and discussions in Appendix L and M of the revised draft. We believe more comprehensive investigation on other domains is beyond the scope of this work, and we leave as future work.
>
> Q2. Clarification on dogs vs. cats experiments
>
> A2. As shown in [3], it is well-known that grayscaling cannot remove all undesired visual biases such as brightness. However, in dogs vs. cats dataset, we found that our randomization technique can outperform other baselines because it makes DNNs capture more meaningful features such as the shape of objects, rather than texture or color. In other words, our randomization method captures an important property by maintaining a spatial (or temporal) structure of input observation. Due to this, policies trained by our methods can be more robust to environments with different observation spaces in many RL benchmarks. We believe that such invariance to low-level transformations in the observation space features is really important in deep RL as R4 mentioned. We clarified this in Section 3.1 of the revised draft.
>
> Q3. Fairness on evaluation
>
> A3. Our inference method is not an ensemble method that requires multiple networks, but a stochastic method based on a single neural network with randomized layers. Hence, we believe that the comparison with the baselines and ours is fair because they are all based on a single neural network, i.e., have the same number of learnable parameters. Furthermore, our methods with a single MC sample (i.e., with no aggregation) still outperform all baseline methods as shown in Figure 3(d) and Table 2.
>
> [1] Wenxuan Zhou, Lerrel Pinto, and Abhinav Gupta. Environment probing interaction policies. In ICLR, 2019.
>
> [2] Charles Packer, Katelyn Gao, Jernej Kos, Philipp Kr ̈ahenb ̈uhl, Vladlen Koltun,  and Dawn Song. Assessing generalization in deep reinforcement learning. arXiv preprint arXiv:1810.12282, 2018
>
> [3] Byungju Kim, Hyunwoo Kim, Kyungsu Kim, Sungjin Kim, and Junmo Kim. Learning not to learn: Training deep neural networks with biased data. In CVPR, 2019.

---

### Official Review · AnonReviewer4 · 2019-11-01
**Official Blind Review #4**

**Rating:** 8

**Review:**

This paper proposes applying random convolutions to the observation space to improve the ability of deep RL agents to generalize to unseen environments. To encourage the learning of invariant features, the authors further include a loss term to align features of perturbed and unperturbed observations. Thorough experiments on multiple generalization benchmarks show that this method outperforms many previously used regularization and data augmentation techniques.

Although the proposed method is simple, it represents a useful contribution. The need to generalize across low level transformations in the observation space features prominently in several environments, including DeepMind Lab and CoinRun. The clear need for agents to be invariant to these low level transformations well motivates the proposed approach, as does the failure of many existing methods to provide this invariance.

The authors could more explicitly discuss the main drawbacks of this approach. As with any data augmentation, there is an assumption that the applied transformation generally won’t destroy information pertinent to the task. While this is true for the MDPs investigated here, it is easy to imagine slight variants of these MDPs for which this approach would fail. If an optimal policy must condition on color or texture information from observations, then using these random convolutions would render training impossible. Encountering such MDPs is not farfetched, so this weakness seems worth acknowledging.

In Figure 5 it would be useful to visualize performance of an agent trained directly on these unseen environments, as this presumably serves as an upper bound for the zero-shot performance of “PPO + ours”. How close does “PPO + ours” come to closing this gap? Without any context on the reward scale, it’s hard to infer how well this method is generalizing, beyond seeing that it beats some (possibly weak) baselines. Admittedly some closely related curves can be found in Appendix Figures 9 and 14, though they’re a bit out of the way.

Section 3.1 mentions that using alpha = 0 complicates training. It is somewhat surprising that using alpha > 0 is necessary or significant and yet the value used (alpha = .1) is relatively small. Any further comments on this choice?

I appreciate the discussion in Appendix F. It’s natural to wonder about alternative injection sites for the random network, and it’s good to see how the proposed method compares to these alternatives.


**Experience Assessment:**

I have published one or two papers in this area.

**Review Assessment: Checking Correctness Of Derivations And Theory:**

N/A

**Review Assessment: Checking Correctness Of Experiments:**

I carefully checked the experiments.

**Review Assessment: Thoroughness In Paper Reading:**

I read the paper thoroughly.

---

> ### Author Response · Authors · 2019-11-15
> **Response to R4**
>
> We appreciate your valuable comments, efforts and times on our paper. As you and R3 mentioned, we introduce a simple, yet powerful solution for generalization in deep RL, and show that it achieves significant performance gains in comparison to many known regularization and data augmentation techniques in various environments. Our responses to all your questions are provided below. Revised parts in the new draft are colored by red (in particular, we updated Section 3, 4 and 5, Appendix A, L, M, and N, and Figure 5, 7, 16, and 17).
>
> Q1. Handling potential failure case of our methods
>
> A1. Thank you for the suggestion to acknowledge the failure cases of our method. As you pointed out, some color (or texture)-conditioned RL tasks can be difficult for our methods to work. For example, consider an extreme seek-avoid object gathering setup, where the agent must learn to collect good objects and avoid bad objects which have the same shape but different colors. However, we remark that our method would not always fail for such tasks if other environmental factors (e.g., the shape of objects in Collect Good Objects in DeepMind Lab [1]) are available to distinguish them. As shown in Appendix M of the revised draft, our method can perform well on the modified CoinRun environment with good and bad coins by capturing the other factors such as the location of coins to perform this task. As another example, consider color-matching tasks such as the keys doors puzzle in DeepMind Lab [1] where the agent must collect colored keys to open matching doors. Even though this task is color-conditioned, a policy trained with our method can perform well because the same colored objects will have the same color value even after randomization, i.e., our randomization method still maintains the structure of input observation. Finally, we remark that our method can handle such corner cases by adjusting the fraction of clean samples during training ($\alpha$ on page 4, in the paragraph "Details of the random networks."). In summary, we believe that the proposed method covers a broad scope of generalization across low-level transformations in the observation space features as you mentioned. We added related discussions in Section 5 and Appendix M of the revised draft.
>
> Q2. “Optimal” performance (i.e. upper bound)
>
> A2. Following your suggestion, we included the performance of an agent trained directly on unseen environments to Figure 5 and 7of the revised draft. In summary, our method approaches this hypothetical upper bound performance.
>
> Q3. The choice of $\alpha$
>
> A3. We would like to clarify that the chosen fraction $\alpha=0.1$ of clean samples is set empirically based on controlled experiments. To support this, we added controlled experimental results by varying the fraction of clean samples in Figure 17(d) of the revised draft.
>
> [1] Charles Beattie et al. Deepmind lab. arXiv preprint arXiv:1612.03801, 2016.

---

### Author Response · Authors · 2019-11-15
**Common response to all reviewers: short summary of rebuttal**

Dear reviewers,

First of all, we appreciate your efforts in providing valuable comments on our manuscript.

To best respond to the questions and concerns raised by the reviewers, we have carefully revised and enhanced our manuscript with a substantial amount of additional experiments following suggestions from the review comments, including:
- Adding oracle performances of PPO agents (Section 3 and 4)
- Ablation study (Appendix N)
- Discussion on potential failure cases of the proposed method (Appendix M)
- Extension of the proposed method to domains with different dynamics (Appendix L)
- More clarification on dogs vs. cats experiments (Section 3)

Revised parts in the new draft are highlighted in red (in particular, we updated Section 3, 4 and 5, Appendix A, L, M, and N, and Figure 5, 7, 16, and 17).

Thank you very much.
Authors.

---

### Decision · Program_Chairs · 2019-12-19

**Decision:**

Accept (Poster)

**Comment:**

This submission proposes an RL method for learning policies that generalize better in novel visual environments. The authors propose to introduce some noise in the feature space rather than in the input space as is typically done for visual inputs. They also propose an alignment loss term to enforce invariance to the random perturbation.

Reviewers agreed that the experimental results were extensive and that the proposed method is novel and works well.

One reviewer felt that the experiments didn’t sufficiently demonstrate invariance to additional potential domain shifts. AC believes that additional experiments to probe this would indeed be interesting but that the demonstrated improvements when compared to existing image perturbation methods and existing regularization methods is sufficient experimental justification of the usefulness of the approach.

Two reviewers felt that the method should be more extensively compared to “data augmentation” methods for computer vision tasks. AC believes that the proposed method is not only a data augmentation method given that the added loss tries to enforce representation invariance to perturbations as well. As such comparisons to feature adaptation techniques to tackle domain shift would be appropriate but it is reasonable to consider this line of comparison beyond the scope of this particular work.

Ac agrees with the majority opinion that the submission should be accepted.